# Optimizing success rate with Nonlinear Mapping Control in a high-performance raspberry Pi-based light source target tracking system

**Guiyu Zhou[1], Bo Zhang[1,2], Qinghao Li[3], Qin Zhao[4], Shengyao Zhang**[5]*

**1** School of Electronic Information and Engineering, Yibin University, Yibin, China, **2** Shanghai Judong Semiconductor Company Limited, Shanghai, China, **3** School of of Mechanical and electrical engineering, Yibin University, Yibin, China, **4** School of Business English (Yibin Campus), Chengdu International Studies University, Yibin, China, **5** School of Mathematics and Physics, Yibin University, Yibin, China

* zsy8224839@163.com

## Abstract

This study addresses the limitations of linear mapping in two-dimensional gimbal control for moving target tracking, which results in significant control errors and slow response times. To overcome these issues, we propose a nonlinear mapping control method that enhances the success rate of light source target tracking systems. Using Raspberry Pi 4B and OpenCV, the control system performs real-time recognition of rectangular frames and laser spot images. The tracking system, which includes an OpenMV H7 Plus camera, captures and processes the laser spot path. Both systems are connected to an STM-32F407ZGT6 microcontroller to drive a 42-step stepper motor with precise control. By adjusting the parameter c of the nonlinear mapping curve, we optimize the system's performance, balancing the response speed and stability. Our results show a significant improvement in control accuracy, with a miss rate of 3.3%, an average error rate of 0.188% at 1.25 m, and a 100% success rate in target tracking. The proposed nonlinear mapping control method offers substantial advancements in real-time tracking and control systems, demonstrating its potential for broader application in intelligent control fields.

## Introduction

In recent years, the integration of automation and intelligence into society has led to the widespread adoption of multidimensional gimbals equipped with automatic tracking systems in both industrial production and daily life [1–3]. Gimbal vision servo technology, which combines control theory, digital image processing, mechanical engineering, and electronic engineering, has emerged as a focal point in current robotics research [4] and has extensive applications across diverse domains [5]. In the security sector, automatic tracking systems intelligently monitor and track targets within surveillance areas, enhancing real-time performance and the effectiveness of monitoring systems [6,7]. This capability provides robust tools for security operations. Similarly, in traffic management, these systems monitor and track traffic flow, thereby increasing the intelligence and automation of traffic systems [8]. This

**Data availability statement:** All relevant data is available on OSF (Open Science Framework): https://osf.io/gpjx5/.

**Funding:** This research was supported by the Key Laboratory of Computational Physics of Sichuan Province (Grant No. 2020JSWLYB001) and Yibin University (Grant No. 2020QH03). The funders had no role in study design, data collection and analysis, decision to publish, or preparation of the manuscript.

**Competing interests:** The authors have declared that no competing interests exist.

is crucial for mitigating congestion and enhancing road safety. In the drone industry, automatic tracking systems enable drones to autonomously follow targets, thereby improving the efficiency of tasks such as aerial photography, monitoring, and search and rescue operations [9,10]. Despite these advances, traditional linear mapping approaches in tracking systems often struggle to achieve quick and accurate control of motor rotation to specified angles. This inadequacy leads to suboptimal real-time performance and considerable tracking errors, underscoring the necessity for innovative methods to improve tracking system capabilities [11–13]. These challenges are particularly pronounced in dynamic and high-speed scenarios where real-time response is critical. Issues such as target occlusion, noise interference, and environmental variability exacerbate tracking errors and limit the reliability of linear mapping methods in practical applications. Additionally, the computational complexity of traditional methods often results in slower system responses, hampering the performance of tracking systems under stringent time constraints. Therefore, addressing these challenges is crucial for improving the effectiveness of automatic tracking systems in real-world settings.

The system consists of two integrated components, the control system and the tracking system, with image processing and servo control playing pivotal roles [14]. Image processing begins with the capture of images via a camera, extraction of target coordinates, and their conversion into deviation data for precise target tracking [15]. Servo control then utilizes this deviation data for real-time closed-loop control of the system. Significant advances have been achieved worldwide in key technologies such as image processing, computer vision, and machine learning [16,17]. For example, the/spl psi/-learning algorithm outperforms traditional support vector machines in scenarios where data are linearly inseparable [18]. It employs a hyperplane tree structure to achieve high classification accuracy, which is particularly effective for handling large and complex objects [19]. In astronomical target tracking, methods such as random forest and deep learning models account for historical observation errors and prediction covariances, ensuring effective target tracking [3,20]. Chinese research institutions and universities are actively contributing to this field, leveraging deep learning and neural networks to enhance image recognition and target tracking capabilities in applications such as video surveillance, autonomous driving, and intelligent transportation. The MK-YOLOv3 algorithm developed by Shenyang Aerospace University is a multiscale clustering convolutional neural network that uses a sampling K-means clustering algorithm with kernel functions to extract image features and determine anchor positions [21]. This approach significantly improves the accuracy and speed in the classification and tracking of small or occluded targets. Various methods are employed in image processing for detecting moving targets, including background subtraction, frame difference, and optical flow techniques [22–24]. Background subtraction compares images against a static background to detect motion, which is suitable for static scenes but susceptible to lighting changes or dynamic backgrounds [23]. Frame difference detects motion by comparing consecutive frames, which is an effective approach for capturing motion but is prone to false positives during rapid motion or camera shaking [25–27]. Optical flow infers motion from pixel displacements between frames and can effectively capture fine-grained motion directions but is challenged by complex backgrounds or occlusions.

In recent years, remarkable advances have been made in the development of servo systems across various fields, including intelligent manufacturing, the internet, and artificial intelligence [28,29]. In typical servo systems, such as those controlling Hard Disk Drive heads, proportion integral derivative (PID) algorithms are commonly employed for precise control [30,31]. Additionally, servo systems that use linear quadratic regulator (LQR) control algorithms have demonstrated enhanced capabilities in controlling multi-input multi-output (MIMO) systems [32]. However, current research often relies on overly complex and costly

vision models, and the integration of vision algorithms with servo systems requires further development and refinement.

This study proposes an innovative method based on nonlinear mapping to improve the success rate of automatic tracking systems. Combining a multi-hardware platform and a nonlinear control strategy, the system can realize efficient and real-time target detection and tracking in a complex dynamic environment. In this process, binary processing and polygon approximation are used to identify the target shape and improve the accuracy and efficiency of the vision algorithm. For the Pan Tilt Zoom (PTZ) control system, the rotation angle and speed must be adjusted with high precision to ensure that the target is always in the field of vision and improve the system stability and response speed. By adjusting the key parameter $C$ value of the exponential function in the nonlinear mapping, the mapping function can adapt to the characteristics of different scenarios and targets to ensure high response speed and accuracy of the control system. The rest of this article is organized as follows: the first section outlines the primary materials and methods used; the second presents the experimental results, including the optimization of the nonlinear mapping parameter and its impact on control accuracy and tracking performance; and finally, we discuss the constraints in laser dot detection, the limitations of this method and the outlook for future work. The detailed abbreviations and definitions used in the paper are listed in Table 1. The results obtained in this study provide theoretical support and technical reference information for target tracking systems in dynamic environments.

## Materials and methods

This system utilizes a solution where the control system and the tracking system operate autonomously. The control system features a red laser as the moving target, while the tracking system uses a green laser for real-time tracking of this target. Both systems function independently without relying on any communication modules. Fig 1 illustrates the setup of the moving target control and automatic tracking system.

### Control system vision scheme

The visual solution for the control system employs a Raspberry Pi 4B and OpenCV (open source computer vision library) for image processing. Initially, the system identifies the 1.8 cm black rectangular target illuminated by the light source. This is achieved through a binarization method followed by polygon approximation. The specific steps are as follows: The raw image collected is first converted to grayscale. To ensure accuracy in the subsequent processing steps, the grayscale image undergoes Gaussian blurring to eliminate noise. After extensive testing, a Gaussian kernel size of 7x7 was determined to provide optimal results for this design. Susequently, the blurred image is converted to a binary image through thresholding. Through testing, the ideal threshold value was established as 55. This means that pixels with a grayscale value above 55 are converted to white, while those with a grayscale

Table 1. List of abbreviation and acronyms used in the paper.

| Abbreviation | Definition | Abbreviation | Definition |
| --- | --- | --- | --- |
| LQR | Linear Quadratic Regulator | MIMO | Multi-Input Multi-Output |
| PTZ | Pan Tilt Zoom | OpenCV | open source computer vision library |
| HSV | Hue, Saturation, Value | OpenMV | Open Machine Vision |
| IDE | Integrated development environment | PID | Proportion-Integral-Derivative |
| GPIO | General Purpose Input/Output | | |

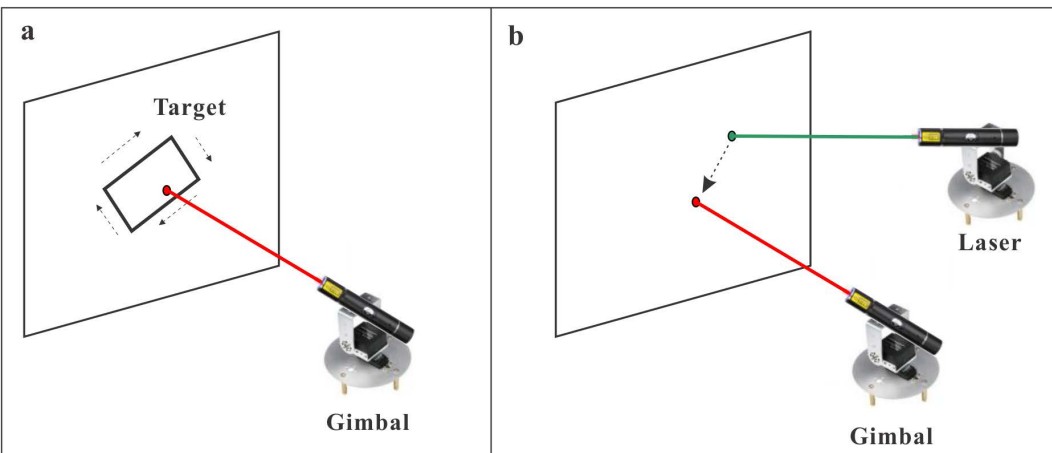

**Fig 1. Motion target control and automatic tracking system.** (a) A control system for moving targets using red laser light. (b) A real-time target tracking system utilizing green lasers for precise and continuous monitoring.

value below 55 are set to black. Since the initial binary image of the black rectangular target appears black and the polygon approximation necessitates white polygons, the binary image must be inverted. This inversion can be carried out in OpenCV using the 'cv2.bitwise_not()' function.

The aforementioned procedures are integral parts of image preprocessing, culminating in the polygon approximation. Initially, the 'cv2.findContours()' function detects all contours in the input image. The contours are subsequently approximated as rectangles using the Douglas-Peucker algorithm. Postapproximation validation ensures that these contours accurately represent rectangles. Notably, multiple rectangles may be approximated within the image, potentially including unwanted interference. Hence, a filtering step is employed based on the value of the rectangle area. Rectangles with the areas ranging from 5000 to 8000 pixels are identified as target rectangles. Finally, the coordinates of the four corners of each target rectangle are extracted and stored in 'rect_corners_list'.

In the control system for laser point recognition, OpenCV plays a crucial role [33]. Initially, the image captured by the camera undergoes conversion to the hue, saturation, value (HSV) colour space. Once this conversion is complete, a specific HSV colour range for red laser points is defined using the 'np.array()' function. The 'cv2.inRange()' function is subsequently employed to filter the region containing red laser points based on the defined HSV colour threshold range. This operation generates a mask where the pixels meeting the criteria are set to 255, while others are set to 0. Following the creation of the mask, morphological operations are applied to further refine the mask image. In this design, a 3x3 kernel is utilized for the erosion 'cv2.erode()' function to eliminate noise from the mask. The dilation 'cv2.dilate()' function is subsequently performed five times to expand the area of the detected laser points, ensuring that the region is appropriately filled.

Following the erosion and dilation operations, the 'cv2.findContours()' function is utilized to detect the contours within the mask image, delineating the red areas corresponding to the laser points. The area of each red laser region is subsequently calculated, and regions with areas of less than 200 pixels are identified as target laser points. The coordinates of these laser points are then extracted, printed, and transmitted via serial communication to the main controller.

## Tracking system vision scheme

In the tracking system utilizing OpenMV (open machine vision) H7 Plus, the detection of red laser points and the measurement of distances between green and red laser points are key functions [34]. The device employs colour recognition to identify red laser points. Initially, specific red threshold values are defined. Using the threshold editor within the OpenMV IDE (integrated development environment), adjustments are made by capturing a frame from the buffer for fine-tuning. The final threshold values are determined by manipulating the Lab colour space values. Fig 2 illustrates the interface of the threshold editor used for this purpose. The specific red threshold values are set as red_threshold = (94, 88, 11, 127, -128, 127).

Upon setting the threshold, the 'find_blobs()' function is employed to detect colour blobs, followed by the use of the 'find_max()' function to identify the largest blob based on area. By utilizing the 'draw_rectangle()' function, a rectangle is drawn around the detected blob, and the 'draw_cross()' function is utilized to mark a crosshair at the blob's centre. The coordinates of the target within the image are then printed. Then, this coordinate information is transmitted to the microcontroller via serial communication.

During the tracking of the red laser point, the green laser and the OpenMV H7 Plus camera are securely mounted together on a gimbal. Error compensation techniques are used to adjust

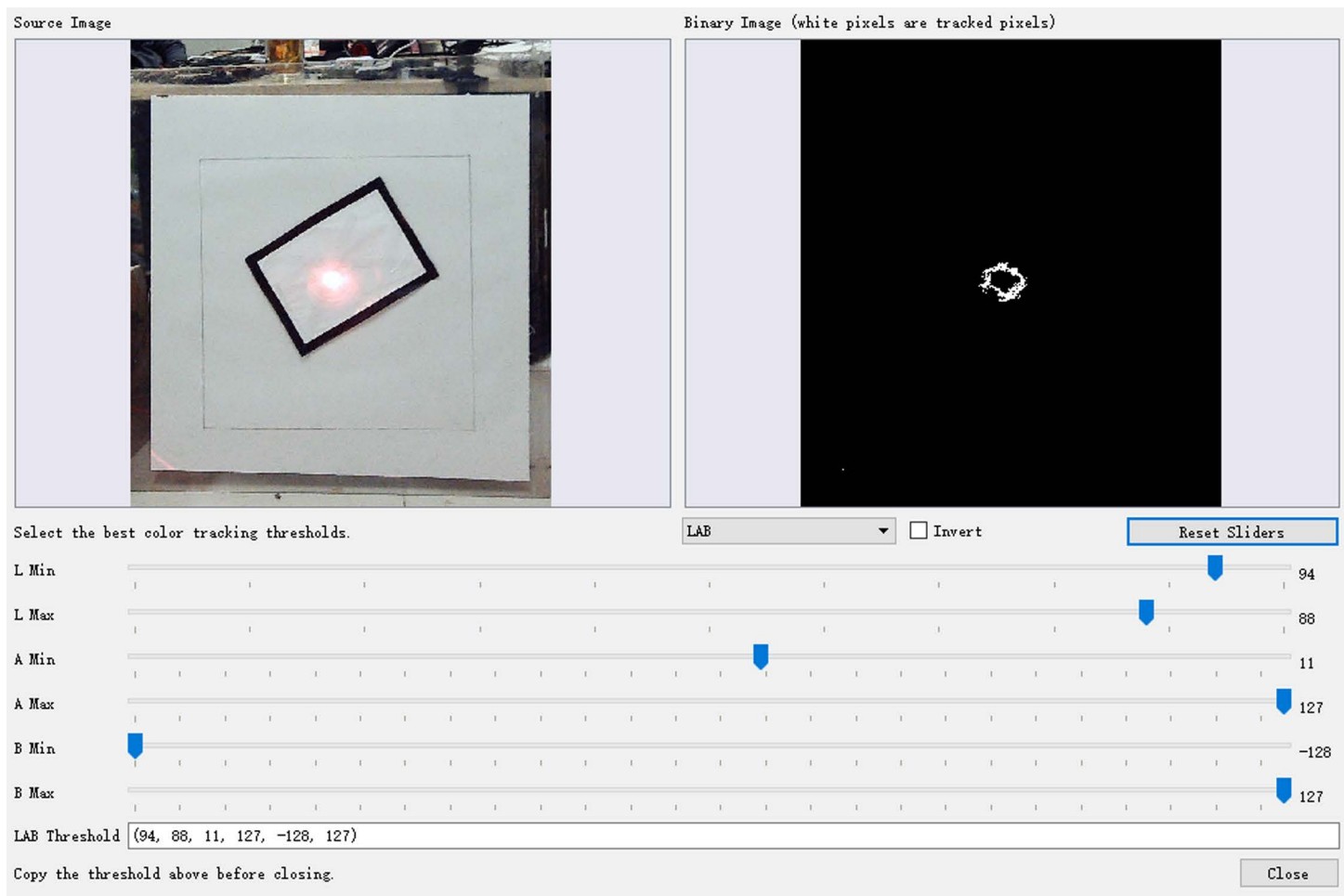

**Fig 2. Threshold Editor.**

the position of the green laser point, ensuring that it remains centred within the field of view of the OpenMV H7 Plus camera. This configuration guarantees that regardless of the gimbal movements, both the laser and the camera maintain relative stability, thereby significantly improving tracking accuracy. As a result, the green laser point consistently remains at the centre of the OpenMV H7 Plus image. The distance between the red laser point and the centre of the OpenMV H7 Plus image serves as the tracked distance.

Upon completing the error compensation calibration, the coordinates of the target object are transmitted via serial communication to the STM32F407ZGT6 microcontroller. The data packet format begins with the header 0xDA and 0xFF to indicate the start of the packet. The coordinates 'x_error' and 'y_error' are represented as two integers, specifying the centre coordinates of the red point. The end of the data packet is marked by 0xAA, 0xAD, 0x8A, and 0x7F as the footer. Data transmission adheres to the little-endian format, where integers are sent in 4-byte segments starting with the least significant byte followed by the most significant byte.

### Nonlinear mapping control scheme

The electrical control gimbal uses the STM32F407ZGT6 core board as its main controller, overseeing 42 stepper motors for precise control. Dual DRV8825 stepper motor driver boards are employed to effectively drive each pair of stepper motors, significantly enhancing control precision. Within the control system, the gimbal features a red laser, while the tracking system's gimbal is equipped with a green laser. Each of the 42 stepper motors requires three control pins: enable, direction, and pulse. Control is straightforward, with enable and direction pins governed by simple high or low logic levels, namely high for enable, low for disable, high for the forward direction, and low for the reverse direction. The pulse pin manages step movements by receiving adjustable frequency pulse signals. A timer configuration regulates GPIO (general purpose input/output) state changes, allowing software-based adjustment of the pulse frequency through a divider. Fig 3 illustrates the control process.

In practical control operations, it is crucial to dynamically adjust the software divider on the basis of real-time errors detected by the camera. Larger errors necessitate a smaller divider to increase motor speeds, whereas smaller errors require a larger divider to reduce motor speeds. This approach ensures optimal performance under varying conditions where traditional PID (Proportion Integral Derivative) control may not achieve precise control. To address this challenge, our design uses nonlinear inverse mapping to achieve accurate control of stepper motor steps. This method allows for fine-tuning motor movements based on the detected error, optimizing performance and responsiveness in the gimbal system.

In this study, an exponential function is employed for nonlinear mapping, enhancing system performance and flexibility. The implementation equation [35] is given by:

$$f(x) = S_{max} + (S_{max} - S_{min}) \times e^{-c\frac{x - D_{min}}{D_{max} - D_{min}}} \tag{1}$$

where $f(x)$ represents the output result of the mapping. $D_{min}$ and $D_{max}$ are determined by the range of the input variables; for example, if the error range of the input is 0–100, $D_{min} = 0$ and $D_{max} = 100$. $S_{min}$ and $S_{max}$ are determined by the desired output range; for example, if the desired output range is 300–3000, $S_{min} = 300$, and $S_{max} = 3000$. The $c$ value is used to control the curve shape of the motor response. It is necessary to flexibly adjust the response speed and accuracy of the motor according to the specific application requirements. A smaller $c$ value results in a slow response, which is suitable for scenarios that require fine control, whereas a larger $c$ value results in a fast response, which is suitable for scenarios that require fast action.

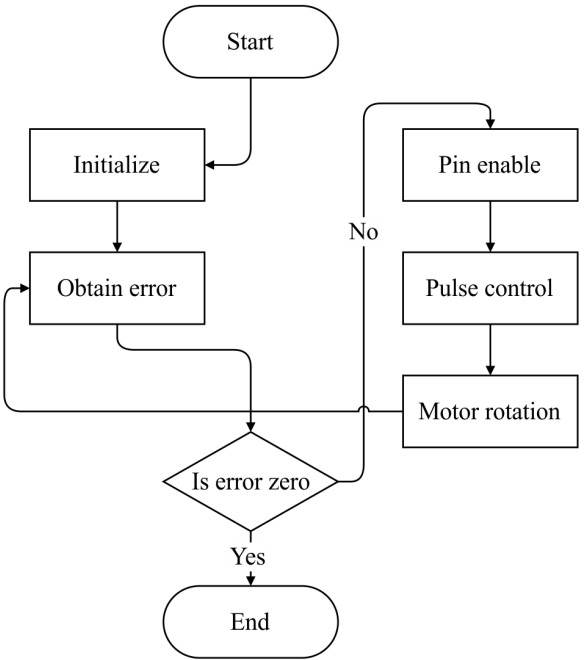

**Fig 3. Motor control flowchart.**

With this function, we generate corresponding output values $f(x)$ based on different input errors $x$. By adjusting the values of $S_{min}$ and $S_{max}$ in the nonlinear mapping function, the range of output values can be controlled. A smaller $S_{min}$ sets the minimum output value of the motor lower and is suitable for scenarios that require a smaller output range and high-precision control. For example, in precision motion control, which requires fine tuning, a lower minimum can help the motor achieve fine adjustment. A larger $S_{max}$ increases the maximum output value of the motor and is suitable for scenarios requiring a larger output range and higher power control. Given the need for fast response or a high load working state, a large maximum can ensure that the motor has enough power to complete the task. By adjusting the exponent $c$ of the nonlinear mapping curve, we control the shape of the motor's response curve. This flexibility allows us to tailor the motor's response speed and precision according to specific application requirements.

## Results

### Nonlinear mapping curve

In the comparison experiment of linear mapping and nonlinear mapping in this study, the hardware components used by the linear system and the nonlinear system, including camera, click, master control, are completely consistent. In addition, the input signal, target dynamic characteristics, noise level and environmental changes are consistent; this ensures that the observed performance differences are due to the mapping method itself rather than external circumstances or data inconsistencies. Using the nonlinear mapping function from Formula 2-1, a MATLAB simulation was conducted with $S_{max}$ set to 8000, $S_{min}$ to 300, $D_{max}$ to 80, and $D_{min}$ to 0. The motor response curves for different values of $c$ are depicted in Fig 4. The simulation results demonstrate the following behaviours:

- For $c = c_1$ or $c_2$, the input range of 0–80 is not fully mapped to the range of 300– to 8000. However, the curve is smooth and decreases gradually, resulting in a gentle system response to errors. This characteristic is suitable for scenarios requiring smooth transitions and where the response speed is not critical.

- For $c = c_3$, the curve shows a moderate descent speed, indicating a faster system response to errors while maintaining stability; this makes it suitable for scenarios where a balance between response speed and stability is needed.

- For $c = c_4$, the output changes very little for inputs greater than 40, with $f(x)$ dropping rapidly thereafter. The curve becomes steep, resembling a step function and resulting in an extremely rapid system response to errors. This characteristic is ideal for scenarios demanding very high response speeds, even though stability may be compromised.

Fig 4 visually demonstrates these distinct behaviours of the motor response curves under different values of $c$, showing how the exponent $c$ influences the motor's responsiveness and stability in practical applications. Based on the simulation results and analysis, a $c$ value of 7 was selected for this design. This choice ensures that the system achieves the necessary response speed while maintaining stability. The balanced response characteristics of this selection enable the system to perform effectively across diverse application scenarios, particularly in control systems that demand both rapid response and high stability.

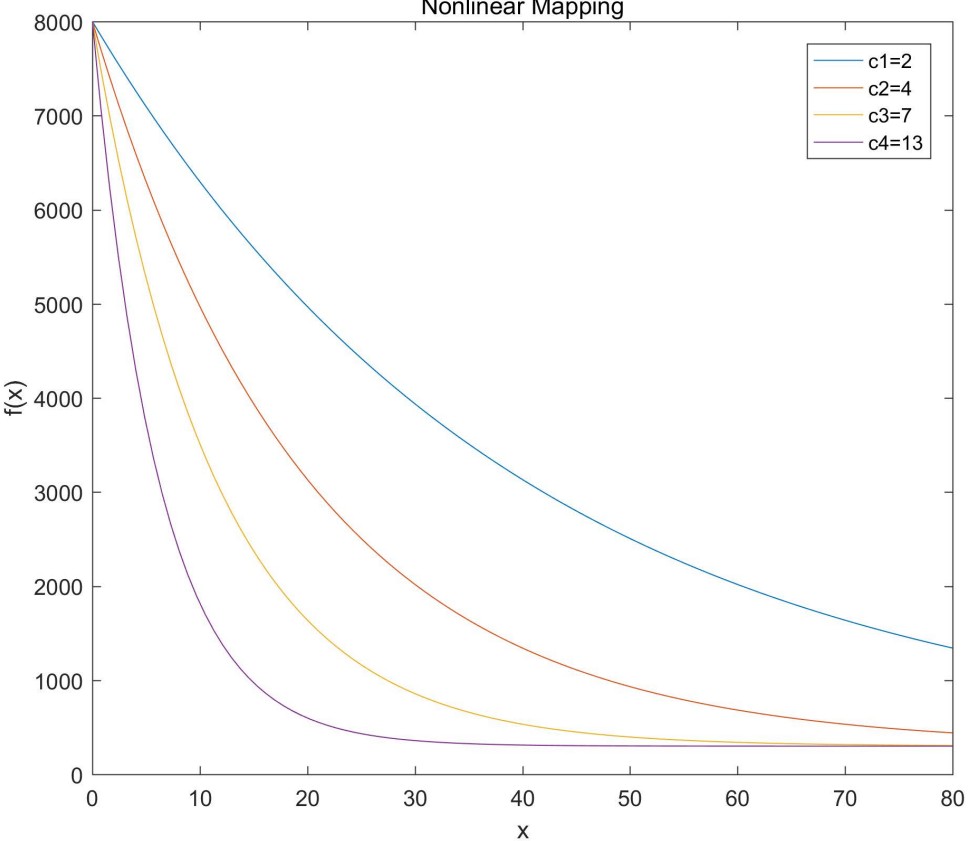

**Fig 4. Motor response curves for different values of c.**

## Control system data testing results

Tables 1 and 2 present the average target recognition times for various target states and testing distances: these values are 1.04898, 1.08979, 1.05726, and 1.10876 s, respectively. Each scenario achieves recognition in approximately 1 s. Fig 5 presents an example of target identification and laser point detection results.

The use of linear mapping to regulate the 42-step motor for laser point manipulation yielded the following performance metrics. At a target distance of 1.25 m from the gimbal, the average operation time was 15.792 s, with a miss rate of 16.7%. At a target distance of 2 m, the average operation time was 15.096 s, with a miss rate of 20%. When the target was tilted, at a target distance of 1.25 m, the average operation time increased to 23.346 s, with a miss rate of 26.7%, a maximum of 2 misses in a single attempt, and a maximum off-target distance of 3.3 cm. At a target distance of 2 m, the average operation time was 20.59 s, with a miss rate of 56.7%, a maximum of 2 misses in a single attempt, and a maximum off-target distance of 3.2 cm. The miss rate increased notably when the target was tilted compared to when it was level. Table 2 lists some test data reflecting these metrics.

Employing a nonlinear mapping function to govern the 42-step motor for laser point manipulation resulted in the following performance metrics: at a target distance of 1.25 m from the gimbal, the average operation time was 14.39 s, with a miss rate of 0%; when the target was extended to 2 m, the average operation time decreased slightly to 14.22 s, accompanied by a miss rate of 3.3%. This reduction in operation time at 2 m is attributed to each step covering a greater distance due to the increased distance, despite maintaining the same step angle.

Under tilted target conditions, at a distance of 1.25 m, the average operation time was 21.23 s, with a miss rate of 6.67%. There was a maximum of one miss in a single attempt, and the maximum distance off-target was 2.7 cm. The miss rate was noticeably higher than that in the level target scenario. At a distance of 2 m, the average operation time was 20.59 s, with a miss rate of 10%. There were up to two misses in a single attempt, and the maximum distance off-target was 1.3 cm.

In the first three scenarios, the system showed satisfactory performance. However, in the fourth scenario, the miss rate was notably higher than those in the other scenarios. Nevertheless, with a maximum of fewer than 5 misses in a single attempt and a maximum off-target distance of less than 5 cm, all of the requirements were met. Table 3 provides a summary of the target recognition time, miss counts, maximum off-target distances, and operation times at the distances of 1.25 m and 2 m for both level and tilted targets.

## Tracking system data testing results

In the detection of the red laser point, a 2.8–12 mm zoom lens is used on the camera to magnify the target at 1.25 m. When running the program offline, the camera achieves processing speeds of up to 49 frames per second, ensuring real-time performance of the tracking system. Fig 6 shows the detection results of the laser point using OpenMV H7 Plus. The tracking time and the distance between two points after completing 20 tests of the tracking system are shown in Table 4.

At a distance of 1.25 m from the target, during the 20 tests using nonlinear mapping, the average tracking time was approximately 0.54 s, with a maximum time of 0.88 s. The average distance between two points was 8.7 mm, with a maximum distance of 17 mm and an average error rate of 0.696%. Occasionally, the two points may coincide, resulting in a distance of 0. By contrast, when linear mapping was used, the average tracking time was approximately 1.69 s,

**Table 2. Motion target control system measurement data(linear mapping).**

| Target Status | Testing Distance (m) | Testing Sequence | Target Recognition Time (ms) | Off-target Occurrences | Maximum Off-target Movement Distance (cm) | Runtime (s) |
|---|---|---|---|---|---|---|
| Horizontal | 1.25 | 1 | 1012.21 | 0 | \ | 15.79 |
| | | 2 | 1074.88 | 0 | \ | 14.23 |
| | | 3 | 1105.42 | 1 | 1.4 | 15.32 |
| | | 4 | 987.08 | 0 | \ | 16.88 |
| | | 5 | 1068.5 | 0 | \ | 14.19 |
| | | 6 | 1007.08 | 0 | \ | 14.06 |
| | | 7 | 1076.98 | 1 | 1.2 | 15.28 |
| | | 8 | 948.74 | 0 | \ | 16.89 |
| | | 9 | 1005.87 | 0 | \ | 15.51 |
| | | 10 | 1052.59 | 0 | \ | 19.77 |
| | 2 | 1 | 1043.73 | 0 | \ | 15.27 |
| | | 2 | 1082.48 | 0 | \ | 14.9 |
| | | 3 | 995.96 | 0 | \ | 14.76 |
| | | 4 | 1038.47 | 1 | 0.5 | 14.81 |
| | | 5 | 1233.32 | 0 | \ | 15.18 |
| | | 6 | 1206.29 | 1 | 1.6 | 15.27 |
| | | 7 | 1156.07 | 0 | \ | 15.27 |
| | | 8 | 1197.82 | 0 | \ | 15.82 |
| | | 9 | 1266.85 | 0 | \ | 14.73 |
| | | 10 | 989.99 | 0 | \ | 14.95 |
| Inclined | 1.25 | 1 | 1047.48 | 0 | \ | 22.51 |
| | | 2 | 1107.74 | 0 | \ | 23.2 |
| | | 3 | 1068.29 | 1 | 2.7 | 26.6 |
| | | 4 | 1036.52 | 0 | \ | 24.12 |
| | | 5 | 994.93 | 2 | 3.3 | 23.38 |
| | | 6 | 1022.42 | 0 | \ | 21.55 |
| | | 7 | 1084.8 | 0 | \ | 22.97 |
| | | 8 | 1049.65 | 0 | \ | 22.92 |
| | | 9 | 1135.62 | 0 | \ | 23.47 |
| | | 10 | 984.16 | 1 | 1.3 | 22.74 |
| | 2 | 1 | 1115.98 | 0 | \ | 20.69 |
| | | 2 | 1121.38 | 0 | \ | 20.4 |
| | | 3 | 1059.02 | 2 | 1.3 | 19.86 |
| | | 4 | 1084.18 | 1 | 0.8 | 20.27 |
| | | 5 | 1073.57 | 0 | \ | 20.23 |
| | | 6 | 1038.28 | 0 | \ | 25.26 |
| | | 7 | 1127.13 | 0 | \ | 19.36 |
| | | 8 | 951.01 | 1 | 0.6 | 20.81 |
| | | 9 | 1019.12 | 1 | 3.2 | 19.54 |
| | | 10 | 1169.21 | 1 | 0.8 | 20.13 |

with an average distance of 19.9 mm and an average error rate of 15.9%. The system maintains continuous tracking within 3 s and consistently keeps tracking errors within 2 cm, meeting standard requirements. During these 20 tests, the success rate for linear mapping was 55%, whereas for nonlinear mapping, it reached 100%.

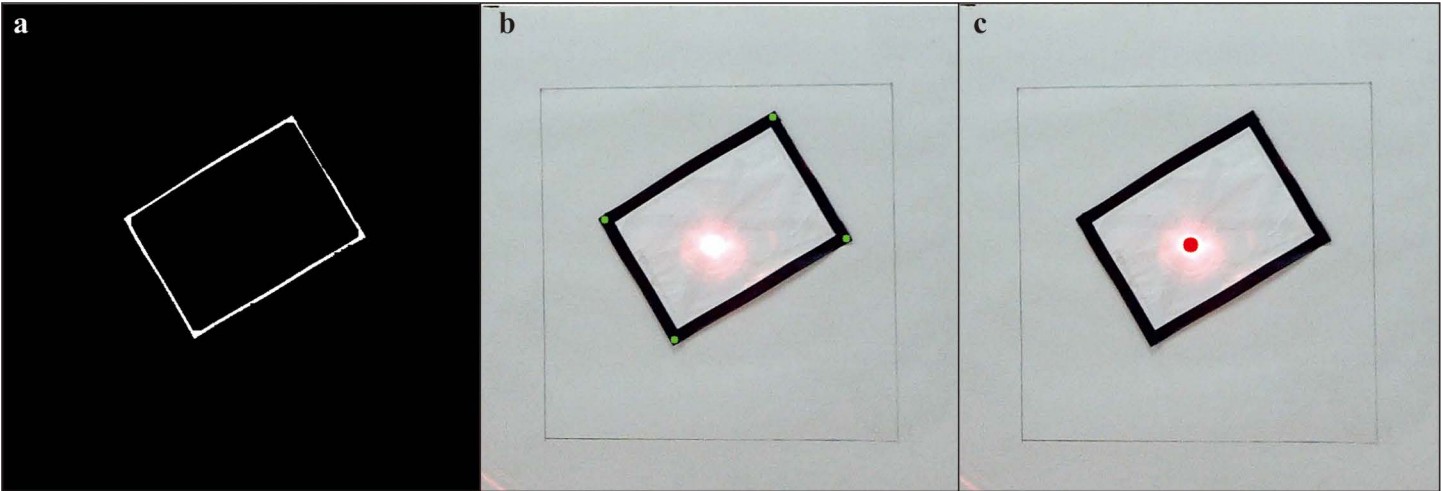

**Fig 5. Target recognition and laser point detection.** (a) The processed result of the rectangular target post binarization and erosion. (b) The outcome of polygonal approximation feature point recognition. (c) The result of laser point recognition.

## Discussion

### Constraints in laser dot detection

After completing rectangle recognition, the coordinates of the four corners of the rectangle are stored in the list 'rect_corners_list', specifically as 'rect_corners_list[0]', 'rect_corners_list[1]', 'rect_corners_list[2]', and 'rect_corners_ list[3]'. When controlling the gimbal, the system starts by sending the coordinates of 'rect_corners_list[0]' and examines whether the red laser dot has reached that position. Once confirmed, the coordinates of 'rect_corners_list[1]' are sent, and the system repeats this process to scan the black rectangular target. However, in practical control, relying solely on the four corners is insufficient for precisely controlling the position of the red laser dot, as it may miss the target along the path between two points [36,37]. Therefore, it is crucial to subdivide the intermediate paths between these four points. In this study, each segment between consecutive points was subdivided into 20 segments; this allows for real-time correction of the red laser dot path, significantly reducing the likelihood of missing the target. The subdivided points are illustrated in Fig 7.

In laser dot detection, there are two main constraints: path subdivision of the control system and the recognition range limit of the tracking system. Path subdivision involves the insertion of n points between the four corner points of the rectangle to achieve precise control. In this study, it is verified that n = 20 can complete accurate control; that is, a distance is divided into 20 parts, and each step is 1/20. The range limit is mainly used to shield interference, and the recognition range can be limited to the target area.

OpenCV uses the HSV colour space for colour recognition to identify the laser dot. However, relying solely on colour recognition can lead to misjudgments [38,39]. When the laser dot is projected onto the target, its high intensity creates an overexposed area at the centre, as observed by the camera [40]. Consequently, any region of high brightness may be erroneously identified as a laser dot, as illustrated in Fig 8a.

To mitigate the impact of interference on the experiment, additional constraints were implemented for filtering in this study. During the target recognition process, the centre coordinates of the rectangle are determined by calculating the midpoint between its diagonal points. The distance between all potential red laser dots and the centre of the rectangle

**Table 3. Motion target control system measurement data (nonlinear mapping).**

| Target Status | Testing Distance (m) | Testing Sequence | Target Recognition Time (ms) | Off-target Occurrences | Maximum Off-target Movement Distance (cm) | Runtime (s) |
|---|---|---|---|---|---|---|
| Horizontal | 1.25 | 1 | 1063.13 | 0 | \ | 16.8 |
| | | 2 | 1019.5 | 0 | \ | 16.67 |
| | | 3 | 1004.26 | 0 | \ | 14.1 |
| | | 4 | 996.66 | 0 | \ | 14.79 |
| | | 5 | 975.99 | 0 | \ | 13.55 |
| | | 6 | 1001.33 | 0 | \ | 13.6 |
| | | 7 | 1143.11 | 0 | \ | 13.47 |
| | | 8 | 1164.54 | 0 | \ | 14.32 |
| | | 9 | 1080.55 | 0 | \ | 13.22 |
| | | 10 | 1040.68 | 0 | \ | 13.4 |
| | 2 | 1 | 1273.16 | 0 | \ | 18.18 |
| | | 2 | 1100.16 | 0 | \ | 12.59 |
| | | 3 | 995.33 | 0 | \ | 14.33 |
| | | 4 | 1038.86 | 0 | \ | 14.24 |
| | | 5 | 1069.69 | 0 | \ | 13.5 |
| | | 6 | 1096.9 | 0 | \ | 13.54 |
| | | 7 | 987.69 | 1 | 0.5 | 13.38 |
| | | 8 | 1148.91 | 0 | \ | 13.65 |
| | | 9 | 1041.28 | 0 | \ | 14.28 |
| | | 10 | 1145.93 | 0 | \ | 14.52 |
| Inclined | 1.25 | 1 | 1014.12 | 0 | \ | 20.42 |
| | | 2 | 1128.24 | 0 | \ | 20.83 |
| | | 3 | 1066.64 | 0 | \ | 19.91 |
| | | 4 | 1042.12 | 0 | \ | 21.24 |
| | | 5 | 1014.27 | 0 | \ | 23.3 |
| | | 6 | 1036.51 | 1 | 1.2 | 21.52 |
| | | 7 | 1095.41 | 0 | \ | 22.89 |
| | | 8 | 1088.66 | 0 | \ | 19.89 |
| | | 9 | 995.11 | 0 | \ | 20.79 |
| | | 10 | 1091.48 | 0 | \ | 21.52 |
| | 2 | 1 | 1214.37 | 0 | \ | 23.32 |
| | | 2 | 1043.17 | 2 | 1.3 | 18.01 |
| | | 3 | 1104.89 | 0 | \ | 18 |
| | | 4 | 1169.21 | 0 | \ | 28.41 |
| | | 5 | 1145.96 | 0 | \ | 23.17 |
| | | 6 | 1132.4 | 0 | \ | 18.69 |
| | | 7 | 1024.94 | 0 | \ | 21.94 |
| | | 8 | 1223.67 | 0 | \ | 18.27 |
| | | 9 | 1051.89 | 0 | \ | 18.23 |
| | | 10 | 977.08 | 0 | \ | 17.82 |

is subsequently computed. Any red laser dot beyond 100 pixels from the centre is considered interference and filtered out. Additionally, the area of the laser dot is calculated, and dots smaller than 10 pixels or larger than 200 pixels are also filtered out as interference. The

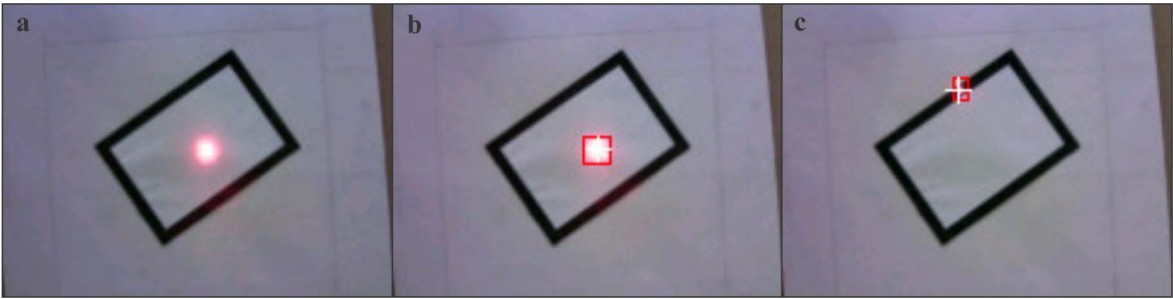

**Fig 6. OpenMV Laser Point Detection Results.** (a) The original unprocessed laser point. (b) The processing results of laser spots projected by the OpenMV H7 Plus on a white background cloth. (c) The processing results of laser spots projected by the OpenMV H7 Plus on a black background cloth.

**Table 4. Tracking system measurement data.**

| Testing Distance (m) | Testing Sequence | Tracking Time (s) | | Point-to-Point Distance (mm) | |
|---|---|---|---|---|---|
| | | Linear mapping | Nonlinear mapping | Linear mapping | Nonlinear mapping |
| 1.25 | 1 | 1.51 | 0.56 | 13 | 0 |
| | 2 | 1.92 | 0.65 | 18 | 11 |
| | 3 | 2.07 | 0.64 | 19 | 9 |
| | 4 | 1.55 | 0.56 | 21 | 8 |
| | 5 | 2.16 | 0.8 | 18 | 8 |
| | 6 | 2.22 | 0.88 | 14 | 4 |
| | 7 | 1.63 | 0.57 | 27 | 13 |
| | 8 | 1.93 | 0.56 | 12 | 3 |
| | 9 | 1.96 | 0.62 | 16 | 2 |
| | 10 | 1.38 | 0.4 | 18 | 5 |
| | 11 | 1.27 | 0.41 | 27 | 15 |
| | 12 | 1.47 | 0.4 | 26 | 12 |
| | 13 | 1.44 | 0.48 | 21 | 7 |
| | 14 | 1.43 | 0.43 | 18 | 10 |
| | 15 | 1.74 | 0.32 | 24 | 14 |
| | 16 | 1.40 | 0.36 | 22 | 15 |
| | 17 | 1.40 | 0.56 | 16 | 8 |
| | 18 | 1.66 | 0.57 | 31 | 17 |
| | 19 | 1.75 | 0.52 | 23 | 11 |

effectiveness of this interference filtering is illustrated in Fig 8b, ensuring stable red laser dot recognition.

## Limitations and outlook

This study successfully designed a motion target control and automatic tracking system utilizing DRV8825 drivers for motor control. However, these drivers are susceptible to overheating [41], particularly under high loads, which may lead to motor or driver damage during prolonged operation. In red laser spot recognition, when the red laser illuminates the black target, most of the light energy is absorbed by the black target, so that the reflected light is weak, and the recognition is inaccurate in some cases. These are the limitations of this study.

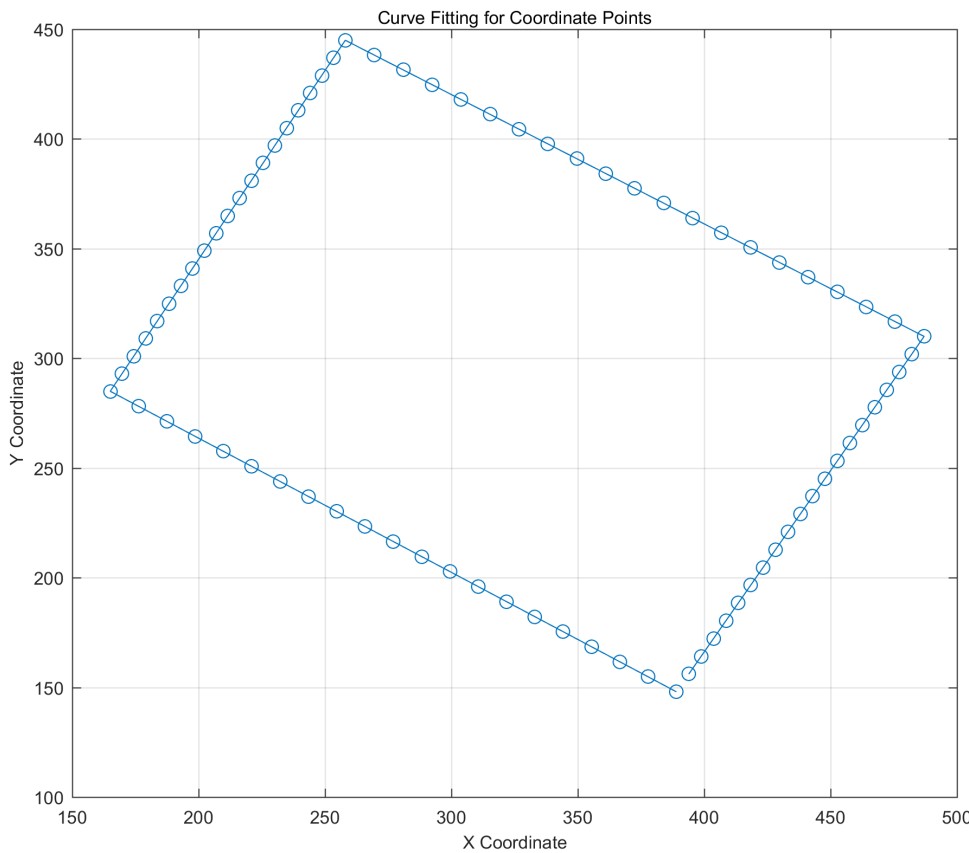

**Fig 7. Subdivided points.**

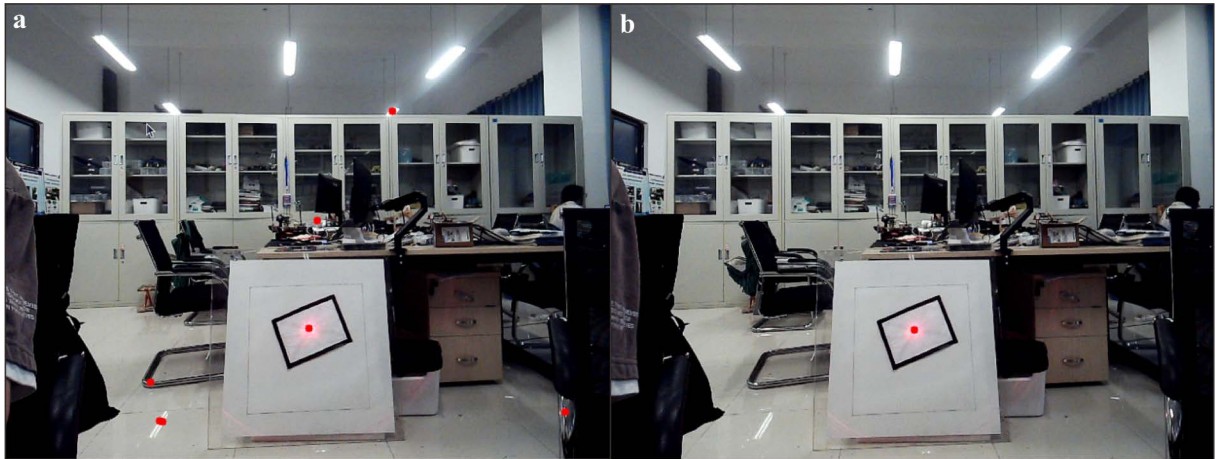

**Fig 8. Interference in the laboratory environment. (a)** (a) Any area of high brightness identified as a potential laser spot. (b) The effective laser points identified after mitigating interference through the application of constraint conditions.

In the future, motor protection can be enhanced by adjusting the drive current or upgrading the driver. System performance can be improved through multithreshold or real-time variable threshold recognition. Future research should focus on the optimization of hardware

design, refining of algorithms, introduction of intelligent control strategies, and enhancement of multisensor fusion to further improve the precision, stability, and response speed of moving target control and tracking systems. In particular, the system's adaptability to complex environments can be significantly enhanced through extensive multiscene testing and adaptive adjustment algorithms. Additionally, integrating artificial intelligence and big data analytics can facilitate intelligent decision-making and real-time feedback mechanisms, enabling the system to better address dynamic practical application scenarios; this will promote its use in broader applications in the military, security, autonomous driving, and other fields.

## Conclusion

The high-performance light source target tracking system based on the Raspberry Pi integrates a control system and a tracking system. The control system uses a Raspberry Pi 4B embedded microcomputer for real-time image acquisition and processing. Computer vision tasks are executed using OpenCV with Python, including functions such as recognizing black rectangular targets and detecting red laser points. The red laser point is controlled by an electric gimbal to manoeuvre on the black target. Target recognition involves methods such as binarization, erosion, contour traversal, and polygon approximation, achieving recognition times of approximately 1 s. Red laser point detection primarily utilizes HSV colour space recognition, complemented by constraints on the rectangular centre and laser point area to filter out interference, thereby ensuring stable recognition. The scanning process around the target is complete within 30 s with high accuracy.

The tracking system employs OpenMV H7 Plus for data acquisition and processing, leveraging MicroPython. Data are transmitted via the serial port to the STM32F407ZGT6 microcontroller, which drives stepper motors with pulse signals to accomplish target tracking. The average tracking time is approximately 0.5 s, maintaining an average distance within 1 cm from the target. When the distance to the target falls below 2 cm, audiovisual alerts are triggered. Both systems utilize open-loop controlled two-dimensional stepper motor gimbals. STM32F4 employs nonlinear mapping control with an optimized parameter value of 7 for $c$, enhancing the system response speed and accuracy. This study underscores the transformative impact of integrating automation and intelligence through multidimensional gimbals equipped with advanced tracking systems. These advances not only improve operational efficiency across various sectors but also pave the way for future innovations in robotics and automation technologies.

## Author contributions

**Conceptualization:** Guiyu Zhou.

**Data curation:** Bo Zhang, Qinghao Li.

**Funding acquisition:** Guiyu Zhou, Shengyao Zhang.

**Methodology:** Qin Zhao, Shengyao Zhang.

**Software:** Qinghao Li.

**Writing – original draft:** Guiyu Zhou.

**Writing – review & editing:** Qin Zhao, Shengyao Zhang.

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
