## [Decision Letter · Decision Letter 0]

26 Nov 2024

PONE-D-24-30172Optimizing Success Rate with Nonlinear Mapping Control in a High-Performance Raspberry Pi-Based Light Source Target Tracking SystemPLOS ONE

Dear Dr. Zhang,

Thank you for submitting your manuscript to PLOS ONE. After careful consideration, we feel that it has merit but does not fully meet PLOS ONE’s publication criteria as it currently stands. Therefore, we invite you to submit a revised version of the manuscript that addresses the points raised during the review process.

We look forward to receiving your revised manuscript.

Kind regards,

Mohsen Bakouri

Academic Editor

PLOS ONE

Journal Requirements:

4. Thank you for stating the following financial disclosure: “Key Laboratory of Computational Physics of Sichuan Province, grant number 2020JSWLYB001 and Yibin University, grant number 2020QH03”.

5. In this instance it seems there may be acceptable restrictions in place that prevent the public sharing of your minimal data. However, in line with our goal of ensuring long-term data availability to all interested researchers, PLOS’ Data Policy states that authors cannot be the sole named individuals responsible for ensuring data access (http://journals.plos.org/plosone/s/data-availability#loc-acceptable-data-sharing-methods). Data requests to a non-author institutional point of contact, such as a data access or ethics committee, helps guarantee long term stability and availability of data. Providing interested researchers with a durable point of contact ensures data will be accessible even if an author changes email addresses, institutions, or becomes unavailable to answer requests. Before we proceed with your manuscript, please also provide non-author contact information (phone/email/hyperlink) for a data access committee, ethics committee, or other institutional body to which data requests may be sent. If no institutional body is available to respond to requests for your minimal data, please consider if there any institutional representatives who did not collaborate in the study, and are not listed as authors on the manuscript, who would be able to hold the data and respond to external requests for data access? If so, please provide their contact information (i.e., email address). Please also provide details on how you will ensure persistent or long-term data storage and availability.

6. PLOS requires an ORCID iD for the corresponding author in Editorial Manager on papers submitted after December 6th, 2016. Please ensure that you have an ORCID iD and that it is validated in Editorial Manager. To do this, go to ‘Update my Information’ (in the upper left-hand corner of the main menu), and click on the Fetch/Validate link next to the ORCID field. This will take you to the ORCID site and allow you to create a new iD or authenticate a pre-existing iD in Editorial Manager.

Additional Editor Comments:

**Please revise your work based on the reviewers comments'**

Reviewers' comments:

Reviewer's Responses to Questions

**Comments to the Author**

1. Is the manuscript technically sound, and do the data support the conclusions?

Reviewer #1: Yes

Reviewer #2: Yes

2. Has the statistical analysis been performed appropriately and rigorously? 

Reviewer #1: Yes

Reviewer #2: Yes

3. Have the authors made all data underlying the findings in their manuscript fully available?

Reviewer #1: Yes

Reviewer #2: Yes

4. Is the manuscript presented in an intelligible fashion and written in standard English?

Reviewer #1: Yes

Reviewer #2: Yes

5. Review Comments to the Author

Reviewer #1: This paper investigates nonlinear mapping control to enhance the success rate of our light source target tracking system. The investigated topic is interesting and the mathematical explanation of the proposed solutions is also nice. Furthermore, the paper is accompanied by a set of simulations for evaluating the effectiveness of the solutions the authors propose. Generally, the paper is well-written and organized. However, some indistinct expressions abate the readability of this paper. Thus, I will recommend several clarifications of this paper which are listed below:

1. The challenges of this work should be discussed in detail in the introduction session.

2. In Introduction, the organization of the paper is needed. The authors should consider these issues in the revision stage.

3. It should be good to strengthen the motivation of this work. The main contributions of this paper are not clear.

4. How do you ensure the comparisons are fair? How are parameters set? Also, how do you ensure the

results are enough to verify the proposal?

5. Are there any constraints in nonlinear mapping control?

6. The quality of presentation is not so good. There are several typos, grammar errors in this manuscript, please check carefully. Also, Figure 1 appears unclear. Please update it in the revision stage.

Reviewer #2: PONE-D-24-30172

Optimizing Success Rate with Nonlinear Mapping Control in a High-Performance Raspberry Pi-based Light Source Target Tracking System

The manuscript should be improved in the revised round according to the following comments.

Comment 1. The abstract is too long, it is recommended to precise the abstract. It is also suggested to include more recent advancements and provide novelty of the proposal.

Comment 2. In this proposal, the parameters Smax, Smin, Dmax, Dmin, and c have a critical role. It is recommended to highlight the impact of these parameters and selection criteria.

Comment 3. The limitation and discussion section should be expanded in the revised draft.

Comment 4. Add some latest studies from the recent literature on the application of the model. For example add Abbas et al. (2024), Abbas et al. (2024), and Abbas et al. (2024).

Comment 5. Provide some future recommendations related to the proposal.

Comment 6. It is recommended to thoroughly read the manuscript and avoid, typos, grammar, and spelling mistakes.

Comment 7. There are many abbreviations, it is recommended to add acronyms in the revised file.

Abbas, Z., Abbas, T., Nazir, H. Z., and Riaz, M. (2024). A novel adaptive CUSUM system for efficient process mean monitoring: An application in piston ring manufacturing process. Alexandria Engineering Journal, 106, 87-100.

Abbas, Z., Nazir, H. Z., Abbasi, S. A., Riaz, M., and Xiang, D. (2024). Efficient and distribution-free charts for monitoring the process location for individual observations. Journal of Statistical Computation and Simulation, 94(13), 2992-3014.

Abbas, Z., Nazir, H. Z., Xiang, D., and Shi, J. (2024). Nonparametric adaptive cumulative sum charting scheme for monitoring process location. Quality and Reliability Engineering International, https://doi.org/10.1002/qre.3522.

6. PLOS authors have the option to publish the peer review history of their article (what does this mean? ). If published, this will include your full peer review and any attached files.

**Do you want your identity to be public for this peer review?** For information about this choice, including consent withdrawal, please see our Privacy Policy .

Reviewer #1: No

Reviewer #2: **Yes: ** Zameer Abbas, Ph.D scholar at School of Statistics, East China Normal University, Shanghai, China.

---

## [Author Response · Author response to Decision Letter 0]

10 Jan 2025

Response to Reviewers

Dear Reviewers:

Thank you for your letter and for the reviewers’ comments concerning our manuscript entitled “Optimizing Success Rate with Nonlinear Mapping Control in a High-Performance Raspberry Pi-Based Light Source Target Tracking System” (ID: PONE-D-24-30172). We are deeply grateful to Reviewer for reviewing the paper so carefully. Those comments are all valuable and very helpful for revising and improving our paper, as well as the important guiding significance to our future researches. We have studied the comments carefully and have made correction which we hope meet with approval. The modified parts in the revised paper have been highlighted with red color. Here are our point-by-point responses:

Reviewer #1’ s comments

Remark #0. This paper investigates nonlinear mapping control to enhance the success rate of our light source target tracking system. The investigated topic is interesting and the mathematical explanation of the proposed solutions is also nice. Furthermore, the paper is accompanied by a set of simulations for evaluating the effectiveness of the solutions the authors propose. Generally, the paper is well-written and organized. However, some indistinct expressions abate the readability of this paper. Thus, I will recommend several clarifications of this paper which are listed below.

Response: Thank you sincerely for your valuable and constructive feedback on our manuscript. We greatly appreciate your recognition of the relevance of our topic, the clarity of the mathematical explanations, and the inclusion of simulations. Your encouraging comments have been instrumental in enhancing the quality and readability of our paper.

We have carefully addressed the points you raised, ensuring that the unclear expressions have been clarified to improve the manuscript’s overall readability. Detailed responses to each of your suggestions are provided below, with the corresponding revisions highlighted in red in the revised manuscript.

Remark #1. The challenges of this work should be discussed in detail in the introduction session.

Response: Thank you for your valuable advice. In response to your suggestion, we have added a detailed discussion of the challenges and difficulties of this research in the introduction section. The challenges and difficulties of the nonlinear mapping automatic tracking system mainly focus on three aspects: first, the accuracy and efficiency of the vision algorithm; second, the stability and response speed of the Pan-Tilt-Zoom (PFZ) control system; and third, the implementation of nonlinear mapping using an exponential function, particularly ensuring that the mapping function adapts to the characteristics of different scenarios and targets. Please see lines 54–62.

Remark #2. In Introduction, the organization of the paper is needed. The authors should consider these issues in the revision stage.

Response: Thank you for highlighting the importance of improving the logical structure of the paper, particularly in the introduction. In response to your suggestion, we have added a clear outline of the main structure of this study in the introduction. Please see line100-103.

Remark #3. It should be good to strengthen the motivation of this work. The main contributions of this paper are not clear.

Response: Thank you for your valuable advice. In response to your suggestion, we have strengthened the discussion on the motivation and clarified the main contributions of this work. The primary contribution and innovation of this research lie in the development of a target tracking method based on nonlinear mapping control. By integrating a multi-hardware platform with a nonlinear control strategy, the system achieves efficient and real-time target detection and tracking in complex dynamic environments. Please see line 60-62 and 93-100.

Remark #4. How do you ensure the comparisons are fair? How are parameters set? Also, how do you ensure the results are enough to verify the proposal?

Response: Thank you for your insightful questions. In the comparison experiment between linear mapping and nonlinear mapping in this study, we ensured fairness by using identical hardware for both systems, including the camera, motor, and main controller, all with the same model and performance specifications. Additionally, the input signals, target dynamic characteristics, noise levels, and environmental conditions were kept consistent. These measures ensure that any observed performance differences are attributable to the mapping method itself, rather than external factors or data inconsistencies, as explained in Lines 214–218.

Regarding parameter settings, these primarily involve the impact of □min, □max, □min, □max, and the range of the index C-value on system accuracy and stability. The specific parameter-setting methods are detailed in Section 4.3 of the paper, with further explanation provided in Lines 196–209.

As for the validity of the results, the test data demonstrate that the system utilizing nonlinear mapping outperforms the linear mapping system in terms of both error reduction and processing time. These results provide robust evidence to verify the effectiveness of the proposed method.

Remark #5. Are there any constraints in nonlinear mapping control?

Response: Thank you for your insightful feedback. In nonlinear mapping control, there are two primary constraints: the path subdivision of the control system and the recognition range limit of the tracking system.

Path subdivision involves dividing the control path into smaller segments to achieve precise control. In this study, it was verified that setting □=20 (dividing the distance into 20 equal parts, with each step corresponding to 1/20 of the total distance) is sufficient for achieving accurate control.

The recognition range limit is primarily used to filter out interference by confining the recognition process to the target area. This ensures the system focuses solely on relevant regions, enhancing its performance in dynamic environments.

These constraints and their implementation have been incorporated into the “Limitations and outlook” Section for further clarification. Please refer to Lines 320–334 for the detailed explanation.

Remark #6. The quality of presentation is not so good. There are several typos, grammar errors in this manuscript, please check carefully. Also, Figure 1 appears unclear. Please update it in the revision stage.

Response: Thank you for pointing out the issues with the quality of presentation in our manuscript. We have carefully reviewed the entire manuscript to address any typos and grammatical errors, ensuring the language and overall readability have been significantly improved. Additionally, we have asked the American Journal Experts (AJE) to review and improve this article. Attached is the certificate, please check.

In the revised version, we have updated Figure 1 with a higher resolution and improved clarity to ensure it is more visually informative and easier to interpret.

We appreciate your valuable suggestions, which have helped us enhance the quality of our manuscript.

Reviewer #2’s comments

Remark #0. The manuscript should be improved in the revised round according to the following comments.

Response: Thank you for your valuable feedback. We appreciate your suggestions and are committed to improving the manuscript in accordance with the comments provided. We have carefully addressed each of the points raised and made the necessary revisions to enhance the quality and clarity of the manuscript. In the following, we address the raised issues in an itemized manner.

Remark #1. The abstract is too long, it is recommended to precise the abstract. It is also suggested to include more recent advancements and provide novelty of the proposal.

Response: Thank you for your valuable suggestion. Based on your feedback, we have revised the abstract to make it more concise and focused on the novelty and recent advancements of our proposed method. We highlighted the key aspects of the nonlinear mapping control and emphasized its improvements in control accuracy and system performance. We believe this revision better reflects the significance and originality of our work. Please refer to the revised abstract in the updated manuscript.

Remark #2. In this proposal, the parameters Smax, Smin, Dmax, Dmin, and c have a critical role. It is recommended to highlight the impact of these parameters and selection criteria.

Response: Thank you for your valuable feedback. We agree that the parameters □max, □min, □max, □min, and □ play a critical role in the precision control of the stepper motor and the tracking process of the light source target. In response to your suggestion, we have now highlighted the impact of these parameters and detailed the selection criteria in the manuscript. Additionally, we have included an analysis of the corresponding high-precision control range for each parameter. Please refer to lines 196-202 for the updated information.

Remark #3. The limitation and discussion section should be expanded in the revised draft.

Response: Thank you for highlighting the need to expand the sections on limitations and discussion. We have specifically added the “Limitations and Outlook” subsection to elaborate on this section, especially in the case where the light source is red and the red laser illuminates a black target. In such cases, most of the light energy is absorbed by the black target, resulting in weak reflected light and occasional inaccuracies in identification. These represent limitations of the current study. In future research, the motor can be protected by adjusting the drive current or altering the drive method, and the system can be enhanced with multi-threshold or real-time variable threshold recognition to improve performance. Please see line 320-334.

Remark #4. Add some latest studies from the recent literature on the application of the model. For example add Abbas et al. (2024), Abbas et al. (2024), and Abbas et al. (2024).

Abbas, Z., Abbas, T., Nazir, H. Z., and Riaz, M. (2024). A novel adaptive CUSUM system for efficient process mean monitoring: An application in piston ring manufacturing process. Alexandria Engineering Journal, 106, 87-100.

Abbas, Z., Nazir, H. Z., Abbasi, S. A., Riaz, M., and Xiang, D. (2024). Efficient and distribution-free charts for monitoring the process location for individual observations. Journal of Statistical Computation and Simulation, 94(13), 2992-3014.

Abbas, Z., Nazir, H. Z., Xiang, D., and Shi, J. (2024). Nonparametric adaptive cumulative sum charting scheme for monitoring process location. Quality and Reliability Engineering International, https://doi.org/10.1002/qre.3522.

Response: Thank you for your valuable suggestion. We appreciate the recommendation to include recent studies in the literature. In response, we have incorporated the suggested references into the revised manuscript to highlight the relevant advancements and applications of the model discussed in our study. Please see the “References”.

Remark #5. Provide some future recommendations related to the proposal.

Response: Thank you for your valuable advice. We have specifically added the “Limitations and Outlook” subsection to clarify future research recommendations. In future research, the precision, stability, and response speed of the moving target control and tracking system can be further enhanced by optimizing hardware design, refining the algorithm, introducing intelligent control strategies, and strengthening multi-sensor fusion. Specifically, to improve the system's adaptability to complex environments, more efficient automatic control can be achieved through multi-scene testing and the development of adaptive adjustment algorithms. Please see line 326-334.

Remark #6. It is recommended to thoroughly read the manuscript and avoid, typos, grammar, and spelling mistakes.

Response: Thank you for your valuable suggestion. We have thoroughly reviewed the manuscript and made the necessary corrections to address the typos, grammar, and spelling mistakes. Additionally, we have asked the American Journal Experts (AJE) to review and improve this article. Attached is the certificate, please check.

Remark #7. There are many abbreviations, it is recommended to add acronyms in the revised file.

Response: Thank you for your valuable feedback. We have carefully reviewed the manuscript and added a list of acronyms to the revised file to improve clarity and readability. Please see line 106.

In summary, under your advice and guidance, we tried our best to improve the manuscript and made some changes in the manuscript. We appreciate for Editors/Reviewers’ warm work earnestly and hope that the correction will meet with approval.

The above responses should answer all the questions. If you have any more questions, please let us know. Once again, thank you very much for your comments and suggestions.

Thank you and best regards.

Very sincerely yours,

Shengyao Zhang

Yibin University

---

## [Decision Letter · Decision Letter 1]

28 Jan 2025

Optimizing Success Rate with Nonlinear Mapping Control in a High-Performance Raspberry Pi-Based Light Source Target Tracking System

PONE-D-24-30172R1

Dear Dr. Zhang,

We’re pleased to inform you that your manuscript has been judged scientifically suitable for publication and will be formally accepted for publication once it meets all outstanding technical requirements.

Kind regards,

Mohsen Bakouri

Academic Editor

PLOS ONE

Additional Editor Comments (optional):

Reviewers' comments:

Reviewer's Responses to Questions

**Comments to the Author**

1. If the authors have adequately addressed your comments raised in a previous round of review and you feel that this manuscript is now acceptable for publication, you may indicate that here to bypass the “Comments to the Author” section, enter your conflict of interest statement in the “Confidential to Editor” section, and submit your "Accept" recommendation.

Reviewer #1: (No Response)

2. Is the manuscript technically sound, and do the data support the conclusions?

Reviewer #1: (No Response)

3. Has the statistical analysis been performed appropriately and rigorously? 

Reviewer #1: (No Response)

4. Have the authors made all data underlying the findings in their manuscript fully available?

Reviewer #1: (No Response)

5. Is the manuscript presented in an intelligible fashion and written in standard English?

Reviewer #1: (No Response)

6. Review Comments to the Author

Reviewer #1: (No Response)

7. PLOS authors have the option to publish the peer review history of their article (what does this mean? ). If published, this will include your full peer review and any attached files.

**Do you want your identity to be public for this peer review?** For information about this choice, including consent withdrawal, please see our Privacy Policy .

Reviewer #1: No

---

## [Editor Report · Acceptance letter]

PONE-D-24-30172R1

PLOS ONE

Dear Dr. Zhang,

I'm pleased to inform you that your manuscript has been deemed suitable for publication in PLOS ONE. Congratulations! Your manuscript is now being handed over to our production team.

Kind regards,

on behalf of

Professor Mohsen Bakouri

Academic Editor

PLOS ONE